# Lateral Formwork Pressure for Self-Compacting Concrete—A Review of Prediction Models and Monitoring Technologies

**DOI:** 10.3390/ma14164767

**Published:** 2021-08-23

**Authors:** Yaser Gamil, Jonny Nilimaa, Mats Emborg, Andrzej Cwirzen

**Affiliations:** 1Building Materials, Department of Civil, Environmental and Natural Resources Engineering, Luleå University of Technology, 97187 Luleå, Sweden; Mats.Emborg@ltu.se (M.E.); andrzej.cwirzen@ltu.se (A.C.); 2Structural Engineering, Department of Civil, Environmental and Natural Resources Engineering, Luleå University of Technology, 97187 Luleå, Sweden; jonny.nilimaa@ltu.se

**Keywords:** self-compacting concrete, form pressure, pressure models, concrete construction

## Abstract

The maximum amount of lateral formwork pressure exerted by self-compacting concrete is essential to design a technically correct, cost-effective, safe, and robust formwork. A common practice of designing formwork is primarily based on using the hydrostatic pressure. However, several studies have proven that the maximum pressure is lower, thus potentially enabling a reduction in the cost of formwork by, for example, optimizing the casting rate. This article reviews the current knowledge regarding formwork pressure, parameters affecting the maximum pressure, prediction models, monitoring technologies and test setups. The currently used pressure predicting models require further improvement to consider several pressures influencing parameters, including parameters related to fresh and mature material properties, mix design and casting methods. This study found that the maximum pressure is significantly affected by the concretes’ structural build-up at rest, which depends on concrete rheology, temperature, hydration rate and setting time. The review indicates a need for more in-depth studies.

## 1. Introduction

Self-Compacting concrete (SCC) is known for providing a convenient working environment, especially in heavily reinforced structural elements. It offers faster construction, in comparison with normal concrete, due to a higher casting rate and having no need for compaction, [1,2,3,4]. Casting fluid-like SCC increases the lateral pressure exerted on the formwork. Commonly, it is assumed to be like the hydrostatic pressure [5]. However, several previous studies indicated a maximum pressure value of approximately 92–95% of the hydrostatic level, and this level was only reached by using a very high casting rate [6,7,8,9,10]. Despite the numerous benefits, SCC is vulnerable to low yield stress increasing the lateral pressure [11,12].

Costs related to formwork are usually high, and studies have shown that the formwork can constitute up to 40–60% of the overall cost of a concrete structure [13,14]. Therefore, overestimating the lateral pressure can lead to additional and unnecessary expenses [15,16,17,18]. The design of the formwork is governed by the amount of lateral pressure exerted while casting. A considerable effort has been made to develop a model to predict these pressures, especially for SCC. While casting SCC, the concrete exerts a horizontal pressure that acts against the surface of the form [19]. The design of formwork is dependent upon the flow-ability, rate of vertical rise, and placing method [20,21]. It is crucial to consider the thixotropic property of the SCC after casting, which is time dependent [22]. Considering that, segregation of SCC can cause a rise in the pressure [23]. The present article reviews crucial issues related to the lateral form pressure developed by SCC. It presents current models and available monitoring technologies and indicates their weaknesses.

## 2. Parameters Affecting the Lateral Formwork Pressure

The design of formwork is governed by the amount of pressure exerted laterally by fresh SCC [12,24]. Parameters controlling that pressure can be classified as concrete mix design, fresh concrete properties and placement technology [12]. Table 1 introduces a list of the parameters affecting the form pressure exerted by SCC.

### 2.1. Concrete Mix Design

Concrete mix design is an important factor affecting the formwork pressure [26,35]. Aggregate grading, cement type, type of chemical admixtures, amount of superplasticizer, and water to cement ratio are certainly some of the most crucial parameters affecting the maximum formwork pressure [12]. Smaller aggregate size leads to higher surface area and reduction of the pressure [37]. A lower sand to coarse aggregate ratio was seen to produce higher thixotropy and thus lower lateral pressure [1,26]. A higher water to cement ratio increases the lateral pressure while a lower water to binder ratio tends to result in less flowable concrete, which decreases the pressure [21,24,27,28,29]. Adding more superplasticizer can diminish the effect of the water to cement ratio. A higher water to binder ratio increases the casting rate which subsequently increases the pressure. In addition, the thixotropy rate is reduced when the water amount increases and that increases the pressure [27,29]. Likewise, the higher content of superplasticizers increases the workability and the lateral pressure [17]. A higher content of fine materials interrupts the ability of coarse material to carry a load of its own weight, thus increasing the pressure [7]. The addition of supplementary cementitious materials (SCMs) such as fly ash and slag cement may also affect the amount of lateral pressure by influencing its rate of pressure decay [7,32]. For example, Saleem et al. [33] indicated that SCC containing fly ash and silica fume showed quicker pressure decay. Processed clays lead to a decrease of the lateral pressure directly after the vertical load is applied due to the fast-structural build-up [62]. Cement type has an impact on the pressure, for example Leemann [38] observed that cement with rapid hardening rates leads to lower pressure after the placement due to fast hydration and development of a self-carrying structure.

Chemical admixtures which make the development of SCC possible have a significant impact on the developing formwork pressure [7]. For example, mixes containing polycarboxylate-based superplasticizers showed slower decrease of the pressure in comparison with naphthalene- and melamine-based admixtures because polynaphthalene sulphonate-based admixture increased the lateral pressure at the initial hydration stage [43]. A higher amount of superplasticizers generally tends to increase the pressure, [43,45]. Retarders tend to lower the rate drop of the formwork pressure while accelerators tend to increase it [29].

### 2.2. Fresh Concrete Properties

Typically, the fresh concrete properties are strongly affected by mix design, i.e., cement type, aggregate grading, chemical admixtures, SCM, etc. The temperature is yet another factor which affects the workability and thus the form pressure by controlling the hydration process, setting times and strength development, [12,39]. Higher temperature leads to lower pressure and faster pressure reduction due to accelerated hydration and thus faster solidification [12,33,39,42]. Longer setting time results in a slower pressure decay [12]. The time required for the pressure to drop has been observed to be equivalent to the final setting time and thus could be determined by a standard penetration test [66]. The pressure appeared also to be sensitive to an increase of the Poisson ratio of concrete, [66]. Adding to that, the density of concrete has a major effect on the pressure amount and both density and pressure have a proportional relationship [41]. Thixotropy is a time-dependent property indicating a loss of concrete fluidity when at rest, which it regains when vibration is applied. It is one of the main factors that affect the form pressure at the initial stage after casting [42]. Addition of Viscosity Modifying Agent (VMA) can change the thixotropy and accelerate the decrease of the pressure [14,19,36,37]. Higher thixotropy leads to a faster pressure decrease [5,14,29]. Higher casting rate decreases the impact of thixotropy on the pressure [57]. During casting, the concrete behaves like any other liquid; it starts to harden and build up internal bonds that can carry the concrete and reduces the formwork pressure [66,67,68].

Low initial shear stress of a fresh SCC due to a low yield stress resulted in a higher pressure in comparison with a normal concrete [17,24]. Some SCMs, i.e., Silica Fume (SF) or Metakaolin (MK) can enhance the shear resistance and thus reduce the formwork pressure [7,36]. Several studies aimed to correlate the formwork pressure with the slump flow, but results were rather inconsistent [7,36,69]. Generally, a high flowable concrete tends to develop a higher pressure which also mainly depends on the casting rate. Figure 1 demonstrates the relationship between flowability and pressure and shows that high flowable concrete generates high lateral pressure. From the findings, the flowability does not alone control the amount of pressure exerted laterally but the casting rate plays a greater role in the pressure than the flowability; if high flowable concrete casts lower, then less pressure is obtained but if a high flowable concrete with high casting rate exists then the pressure is high. Geometry of the form used in the experiments could be the reason for this, as well as the sensors’ accuracy. In fact, if reinforcement is used then there will be some blockage at the sensor diaphragm, causing inaccurate pressure reading. More explanation is needed to address this phenomenon.

### 2.3. Placement Technology

Placement technology and conditions at the construction site strongly affect the formwork pressure. Air humidity and ambient temperature have also been indicated as possible factors affecting the formwork pressure, [52]. Likewise, a higher casting rate tends to increase the pressure [24,28]. In contrast, a low casting rate enables a sufficient structural build-up of the hydrating binder matrix leading to lower formwork pressure [5]. The maximum developed pressure is influenced mainly by the casting rate [24,46]. At a low casting rate, the pressure was observed to decrease even by 50% of the hydrostatic. Higher pressure was observed in mixes with a higher water to cement ratio [21]. Casting rates above 5 m/h resulted in a development of pressures greater than 80% of the hydrostatic pressure [37]. Others indicated that an increase of the casting rate from 5 to 25 m/h generated 15% increase at the initial formwork pressure [58]. Generally, the SCC pressure is lower than the hydrostatic pressure when the casting rate is low. However, it can increase again when late vibration is applied, [46,50]. Example effects of the casting rate on the formwork pressure are shown in Table 2. As observed from Table 2 all the recorded maximum pressure is less than hydrostatic pressure except for casting rate 2.74 m/h because revibrating was applied after casting. The same also happens in the findings by [34,37] because of the high casting rate, which exceeds the hydrostatic pressure. Hence, more studies are required to relate the casting rate and the pressure detection.

Head or vertical rise have also been reported to be significant factors in the variations of pressure level where casting higher structures like walls or columns would generate higher pressure [12,36]. In A study by Ovarlez and Roussel [4], pressure sensors were placed at 0.55 m, 1.95 m, and 3.36 m from the bottom, and it was found that the pressure recorded higher at 3.36 m comparing with the sensors located at 0.55 m and 1.95 m.

Methods of placing and pumping the concrete proved to generate different amounts of pressure. A bottom-up pumping method can generate higher pressure than the hydrostatic pressure due to the additional pressure generated by pumping [37]. Pumping the concrete from the bottom of the structural member is associated with blocking, especially in congested reinforcement and high pressure is required to remove the blocking and disassemble the agglomerates which SCC may build up due to thixotropic behavior and, in consequence of that, the high pressure may damage the form, causing a detrimental effect [57]. The reason for that is the occurrence of hydraulic losses which shows 7% to 16% of the wall pressure [47]. Casting from the top can change the level of thixotropy where the pressure level can be below the hydrostatic pressure [12,37].

Another important factor that has an impact on the formwork pressure is reinforcement, i.e., its orientation and dimensions [54,71]. Congested reinforcement can reduce the pressure by taking over a part of the concrete load. Unfortunately, this effect is negligible for SCC and depends on the maximum aggregate size and the spacing between the bars; hence, more studies are required to expand the understanding of the concrete flow among the reinforcing bars [12].

The type of the material used for formwork can affect the pressure due to variable surface friction, [51]. The rigid and smooth surface tends to increase the pressure similarly as the use of demolding agents by reduction of friction [12]. Steel formwork generated a higher pressure than plywood and watering the surface increased the pressure as well, [59]. The formwork pressure tended to decrease more slowly for steel and PVC surfaces but faster for polyester and plywood [64].

The formwork size was also reported to have an impact on the pressure. Smaller sections tended to generate less pressure due to higher developed friction forces [66]. The circular formworks generated higher pressure than square ones. Other external factors also include weight of equipment, materials, loads form wind and snow [12].

## 3. Modelling

Currently, formwork design assumes the hydrostatic pressure of the concrete as per ACI 347 [72], P = ρgh where P is the pressure in kPa, ρ is the density of concrete in kg/m^3^, g is the gravitational acceleration 9.81 ms^−2^, and h is the casting height (m). However, SCC behaves differently in practice, as it generates pressures lower than the hydrostatic level and there is therefore a possibility to reduce the cost by optimizing the casting rate [19]. This section reviews recently developed models for SCC. A new approach to modeling the formwork pressure for high flowable SCC was developed by Proske and Graubner [73]. The tested concrete had a slump flow of 740 mm and the V-funnel time was either 5 for the low viscosity mix or 13 for the high viscosity mix. The casting rate used was 1, 2, 4, and 8 m/h. The model was developed based on recorded maximum pressure, casting rate, and setting time, see Equation (1).
(1)$$\sigma_{h,E,max}=\frac{\sigma_{hm,max}}{v\times t_{E}\times \rho_{c}\times g}$$
where *σ_h_*_,*max*_ is the pressure plotted against the setting time and casting rate, *v* is the casting rate, *t_E_* is the final setting time, *p_c_* is the density of concrete and *g* is the gravitational acceleration. The maximum pressure for each class based on the final setting time, specific weight, and casting rate can be calculated based on Equation (2).
(2)$$\sigma_{h,max}=0.28\times \nu \times \gamma_{c}\times t_{E}$$
where *v* is the casting rate, *t_E_* is the final setting time, and *γ_c_* is the unit weight of concrete. Unfortunately, the model considers only casting rate and setting time while it neglects other critical factors described earlier.

Another model was proposed by Kwon et al. [46,74]. It considered the effect of intrinsic characteristics of materials but omitted extrinsic effects, i.e., temperature or casting rate [46]. The model assumes that the vertical pressure applied directly after casting is identical to the hydrostatic pressure. By the time the concrete as at rest and started to harden the pressure was assumed to gradually decrease. Two functions were used to predict the ratio of pressure to vertical pressure: *β*(*t*) and *Up* (*t*, *t*′), Equations (3) and (4).
(3)$$\beta \left(t\right)=\beta_{s}-s_{2}\left(t-t_{b}\right)(t>t_{b})$$
(4)$$U_{p}\left(t,t’\right)=U_{p}+\left[\frac{1-U_{p}\left(t,t_{c}\right)}{\left(t_{e}-t_{c}\right)}\left(t-t_{c}\right)\right]\left(t_{c}\leq t\prime \leq t_{e}\right)$$
where *βs* is the value vertical pressure at time tb, *s*_1_, *s*_2_ = initial slope and slope after *tb*, *tb* = time at which slope of *β*(*t*) is changed and it is obtained from the plotted graph where the hydration is noted as the dormant period, *t* = time, *t’* = time during loading, *t_c_* = time where the decreasing rate of *Up*(*t*, *t*′) is suddenly changed. Applying the principle of linear superposition, the pressure was calculated according to Equation (5).
(5)$$\sigma_{l}\left(t\right)=\sum_{i=1}^{N}\Delta \sigma_{V}\left(t_{i}\right)\beta \left(t_{i}\right)\left[1-U_{p}\left(t,t_{i}\right)\right]$$
where $\sigma_{l}\left(t\right)$ is the lateral pressure at a random time t and is expressed in the summation of response $\Delta \sigma_{V}\left(t_{i}\right)$ and each increase in the vertical pressure at time *t_i_*; then, the response is computed. A graphic representation of that model is shown in Figure 2.

Another which can be used for formwork design for SCC has been developed in Germany and included in DIN 18218 standard [75]. The model introduced the pressure measurement for vertical formwork, whereby it focused on different classes of fresh concrete. The maximum nominal pressure *σ_hk_*_,*max*_ was calculated according to Equation (6).
(6)$$\sigma_{hk,max}=\left(1+0.26v\times t_{E\;}\right)\times \gamma_{c\;}\geq 30\;kPa)$$
where *σ_hk_*_,*max*_ is the maximum pressure exerted by fresh SCC, *γ_c_* is the unit weight of fresh concrete, *t_E_* is the setting time of concrete, and υ is the mean casting rate. Figure 3 shows the maximum pressure developed in different classes of consistency vs. the casting rate per hour. The model assumes a concrete weight of 25 KN/m^3^, limits the casting time to the range between 5 and 20 h, sets the temperature at 15 °C and the casing rate to 7 m/h and limits the slump flow of concrete to 600 mm. It assumes usage of internal vibrators while casting and that the casting height is <10 m. Consequently, the model has only a very limited applicability.

The model by Khayat and Omran [68] was based on data obtained from their self-developed portable pressure column. The setup enabled monitoring of the formwork pressure on a small scale using polyvinyl chloride (PVC) with the dimensions of 700 mm height and 200 mm diameter. Two pressure sensors were mounted at 63.5 mm from the bottom and 63.5 mm from the top. The pressure was then monitored at a different time interval and the hydrostatic pressure was compared with the lateral pressure exerted by the mix which had a slump of 660 mm. The ratio of *P_hydro_* to *P_max_* was compared with the height of the concrete. The results showed that the lateral pressure was less than the hydrostatic pressure. The developed model used the relation between the hydrostatic pressure and the exact pressure exerted by SCC using the following ratio of K_0_ = P_max_/P_hydro_ with the numerical data obtained from the rheological analysis of stress and the model then introduced in Equation (7):(7)$$K_{0}=[112.5-3.8h+0.63R-0.6T+10D_{min}-0.021PV\tau_{0rest@15min}]\times f_{MSA}\times f_{Wp}$$
where *h* is the height, *R* is the casting rate, *T* is the concrete temperature in °C, *Dmin* is the formwork dimension, and *PVτ*_0_*_rest@_*_15*min*_ is the static yield stress which is obtained from vane test. *f_MSA_* and *f_Wp_* are factors of safely representing maximum size aggregate and the effect of time, respectively. The value of 1.0 is considered for both factors except for 1.10 for MSA in the case of small coarse aggregate approximately 10 mm and casting of 12 m height and *f_Wp_* decreases to 0.9 for placement with 30 min rest period. The concrete temperature is 22 °C.

Another model developed by Graubner et al. [42] considered material properties using a semi-probabilistic safety concept and soil mechanics. The pore pressure measurements were related to the change in height h and the maximum pressure *σ_h,max_* was assumed to be at the maximum height *h_max_*. The subsequent reduction of pressure was assumed to be caused by the structural build-up. The decrease in pore water content was assumed to be related to the thixotropic build-up and hydration process. Initially, Graubner et al. [64,73] used the following formula to calculate the maximum horizontal pressure, Equation (8).
(8)$${\bar{\sigma }}_{h,E,max}=\left(\frac{\sigma_{h,max}}{\sigma_{h,E,hydro\;}}\right)$$
where ${\bar{\sigma }}_{h,E,max}$ is the normalized maximum pressure corresponding to the maximum horizontal pressure *σ_h,max_* divided by the hydrostatic pressure *σ_h_*_,*E*,*hydro*_
(9)$$\sigma_{h,max}={\bar{\sigma }}_{h,E,max}\times v\times t_{E\;}\times \gamma_{c\;}$$
where v is the casing rate, t_E_ is the setting time measured using the Vicat penetration test, *γc = ρc* ∗ *g*, with concrete density *ρc* and gravity constant *g* and ${\bar{\sigma }}_{h,E,max}$ is calculated for SCC using Equation (10):(10)$${\bar{\sigma }}_{h,E,max}=0.16+\frac{0.8}{h_{E}}\;\leq 1$$
where *h_E_* is the height measured from the level of the hardened concrete to the location of the concrete pump. Figure 4 shows the differences between different classes of concrete with respect to the pressure and height. As a conclusion of the model, the only parameters that were included are the casting rate, setting time, and density of concrete.

Gardner et al. [23] suggested yet another model to predict the lateral pressure exerted by SCC on the formwork pressure but this time based on field measurements. The used concept predicted time after casting when the slump of the placed concrete decreased to zero (*t*_0_). However, since this measurement was not possible in practice, instead a value of 400 mm (*t*_400_) was set as the indicator, Equation (11).
(11)$$t_{0}=t_{400}\left[\frac{Initial\;slump\;flow}{initial\;slump\;flow-400\;mm}\right]$$

The maximum lateral formwork pressure was developed from the equation, see Equation (12), where w is the unit weight of concrete, *R* is the casting rate, *t*_0_ is the initial slump flow value from Equation (12).
(12)$$P_{max}=\frac{wRt_{0}}{2}$$

Teixeira et al. [76] used an empirical model developed by Santilli and Puente [40] but added several parameters, i.e., slump flow, concrete temperature, placement rate, cement type, the height of casting, and the minimum and maximum size of the cross-section, Equation (13).
P_max_ = Kγh(13)
where P_max_ is the maximum lateral pressure against the formwork, K is the reduction coefficient, γ is the specific weight of concrete and h is the height of the concrete. The reduction factor considers seven factors as in Equation (14):(14)$$K=K_{R}K_{\alpha }K_{H}K_{T}K_{d}K_{c}K_{ST}$$
where $K_{R}$ is the coefficient of correction for casting rate, $K_{\alpha }$ represents slump flow, $K_{H}$ is the height of concrete, $K_{T}$ is the concrete temperature, $K_{d}$ is the minimum dimension of cross-section, $K_{c}$ is the cement type, and $K_{ST}$ is the cross-section type. These factors were determined using regression analysis for the measured pressure and the pressure calculated considering the factors as coefficient K factors one by one. Teixeira et al. [76] assumed a bilinear distribution which is not always the case due to the thixotropic behavior of the concrete. The model also omitted several crucial factors, i.e., concrete behavior or structural build-up.

The last model considered here, and the newest model is the one introduced by Assaad and Matar [56] who proposed a regression model. The model considered the presence of transverse and vertical reinforcing bars. The model is based on experimental data from 32 different SCC mixes in 1.6 m formwork 1.6 m high. Studied concretes contained recycled concrete aggregate (RCA). Results indicated that RCA tended to reduce the initial maximum pressure due to a higher surface roughness which increased the internal friction and material build-up at rest. The presence of steel bars confined the plastic concrete and retained part of the load. The pressure decay was controlled by different factors, i.e., rate of hydration, friction, amount of RCA, and thixotropy. It was also found that horizontal bars more strongly reduced the formwork pressure than vertical bars. The model was formulated as shown in Equation (15)
(15)$$\sigma_{\frac{max}{hydro},\%=-23A_{Thix}-2.6Q_{w}-1.45\;eff\left(\rho_{sv}\right)-16\;Eff\left(\rho_{st}\right)+103.8}$$
where *A_Thix_*, *Q_w_*, *eff_(_*ρ*_sv)_*, *and eff_(_*ρ*_st_*_)_ are the thixotropy, the relative water absorption factor, vertical steel density index, and transverse steel density index, respectively.

A physical pressure prediction model developed by Ovarlez and Roussel [4] based on rheological properties of SCC included the apparent yield stress. The model considered the resting zones which differ based on the cross section and dimension. Therefore, two models were developed to fit different cross sections. In the case of a rectangular formwork, Equation (16) is used to determine the pressure and Equation (17) for the case of columns.
(16)$$\sigma \left(rectangular\right)=Kh(\rho g-\frac{hA_{thix}}{eR})$$
(17)$$\sigma \;\left(circular\right)=Kh\left(\rho g-\frac{hA_{thix}}{rR}\right)$$
where *σ* denotes the lateral stress (pressure), *K* is 0.97 and depends on the amount of air entrapped within the concrete, *h* is the depth, *ρ* is the density, *g* is the gravitational force, *A_thix_* is the thixotropy (flocculation coefficient measured by rheometer), *e* is the width of the concrete element, *r* is the radius in the case of circular column, and *R* is the casting rate.

An overview on the main influencing parameters considered in reviewed models is shown in Table 3. Most considered were the density of concrete, casting rate, and temperature. On the other hand, the effect of aggregate size and binder properties were not always considered despite their proven impact.

Beside the models discussed previously, there are also other studies that focused on the topic of lateral form pressure exerted by SCC. Table 4 highlights an overview of these studies to produce a clear pathway for further improvement in future studies.

## 4. Formwork Pressure-Monitoring and Measurements

The formwork pressure was monitored using various approaches and different types of sensors; Table 5 demonstrates different types of measurement tools. Examples include pressure transducers, strain gauge-based pressure sensors, and flush diaphragm millivolt output type pressure sensors [13,37,76,78]. Sensors can be flush mounted on the formwork surface, or on horizontal structural members of the supporting system to measure developing strains.

The following discussion introduces several used laboratory and full-scale setups. For example, Kwon et al. [74] developed a laboratory setup to monitor the lateral pressure and established a special apparatus that can exclude the extrinsic factors, see Figure 5. The apparatus was made of steel to avoid the effect of formwork flexibility and the cylinder diameter was 150 mm and its height 350 mm and grease was applied on the surface as a demoulding agent. During the experiment, the concrete was filled to 300 mm height, an air compressor was inserted through the top plate of the apparatus, and the air pressure was controlled by the pressure gauge. The test was performed indoors at 20 °C. Two pressure cells with 175 kPa capacity were used to measure the lateral pressure and placed at the middle height and on the opposite side of the cylinder. Two different loading cases were used to determine the amount of pressure. One when a vertical pressure was applied at various times after casting and sustained over time. The second included application of a vertical pressure but was increased stepwise.

McCarthy and Silfwerbrand [80] used three approaches to monitor the pressure, i.e., using direct pressure sensors, measuring tension force at ties, and measuring strain in formwork members, see Figure 6. The lateral pressure was measured using flush-mounted pressure sensors.

Another laboratory setup used a column made of transparent Plexiglas acrylic formwork [76]. The formwork had a height of 1600 mm, length of 400 mm, and width of 200 mm. The transparent acrylic enabled visual observations of casted concrete. Additionally, two vertical reinforcing bars with diameters of 20 mm were installed, having transverse links with diameters of 10 mm. Three pressure sensors were installed at 100 mm, measured from the bottom of the formwork. A similar setup but using a PVC pipe and without reinforcement was used in another study [71,74,81]. The formwork could sustain up to 600 kPa. The sensors were located at 63 mm from each end. Perrot et al. [19] developed a laboratory setup for a 2 m high column and attached pressure transducers, while the diameter was 200 mm. The formwork was made of PVC and the pressure was monitored using pressure transducers which were bolted directly into the column at 200 mm and 400 mm height from the base. This investigation aimed to monitor the pressure at a low casting rate varying between 0.55 and 2.5 m/h.

Santilli et al. [39,40] used a metal formwork where the pressure diaphragm sensor was welded on the surface of the formwork, see Figure 7. The formwork was rectangular, 25 m × 25 cm and had a height of 1.2 m and a sheet thickness of 3 mm. Sensors were mounted 100 mm from the bottom of the formwork and the sensor diameter was 19 mm. Casting of concrete was done from the bottom and the pressure was monitored using installed pressure sensors.

Yet another example of a small-scale laboratory setup was developed by Benaicha et al. [11], see Figure 8. It used flush-mounted diaphragm pressure sensors. A square column form had dimensions of 16 × 16 × 70 cm. Four sensors were used.

It is observed that more improvement is required to address the uncovered parameters affecting the amount of lateral pressure induced by SCC. It is still a challenge and a need for designers of formwork to accurately estimate the formwork pressure and current practice is based on hydrostatic, concerning which, in reality, concrete does not behave as any other fluids due to the presence of the thixotropic property and the time-dependent structural build-up. As a result, improper estimation of the formwork pressure is accompanied by an undesirable outcome in terms of compromising safety and leads to unnecessary cost.

## 5. Conclusions

The form pressure developed by fresh SCC has received significant attention due to the increasing usage of this type of concrete. The development of a reliable system to monitor and predict this pressure requires a very good understanding of the controlling factors and the involved mechanisms. So far, several prediction models have been developed but none is sufficiently accurate and universal. Most models assume the lateral pressure to increase linearly with depth and to reach a maximum value which then remains constant. However, the pressure change is a time-dependent phenomenon. Furthermore, the lateral force has been measured to generally reach only maximum pressure of approximately 90% of the hydrostatic pressure during casting. After casting, once, the concrete mix stabilizes and the thixotropic behavior of the concrete starts to develop a self-carrying structure, which reduces the concrete pressure in bottom layers. Furthermore, the hydration processes rapidly initiates a build-up of the binder matrix, consisting of loadbearing ettringite and C-S-H at this stage, thus reducing the pressure. In general, current models do not sufficiently consider basic materials properties, which is especially important for ecological SCC incorporating various types of SCMs that affect the fresh concrete properties and early strength development. A fully developed prediction model should also consider the temperature effect, which is well known to control hydration processes. The reliability of most sensors currently used to monitor the formwork pressure is strongly affected by the build-up of a solid binder matrix or thixotropy of the concrete mix and it is therefore important that used sensors can capture the true pressure level also during the hardening phase.

In summary, determination and prediction of the lateral formwork pressure exerted by SCC require further research, including on the effects related to material properties, mix design, placement techniques and casting rates, rheology, temperature, setting times, hydration rate, stiffness build-up and sensors and their installation and data interpretation, and modelling.

## Figures and Tables

**Figure 1 materials-14-04767-f001:**
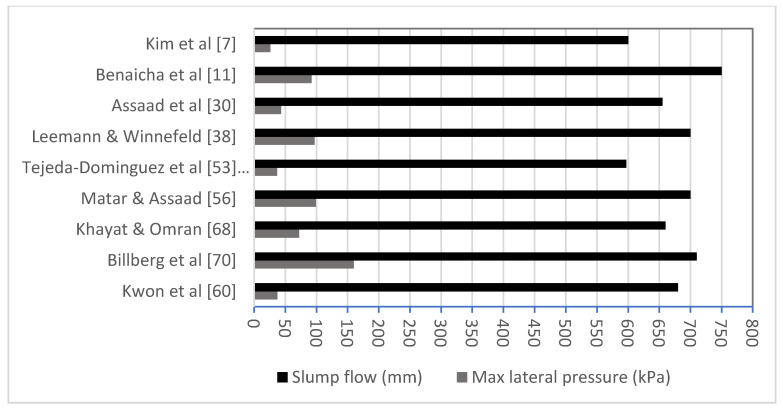
Correlation between slump flow and formwork pressure for different mix designs.

**Figure 2 materials-14-04767-f002:**
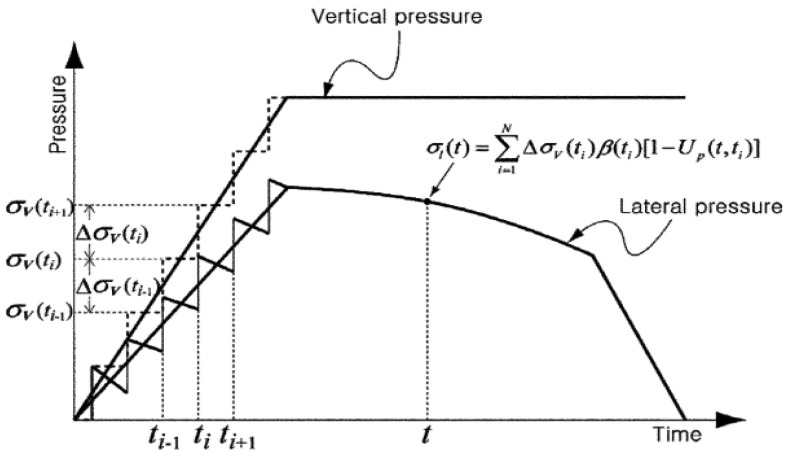
Lateral pressure calculation method [46] (reprinted with permission from © ACI).

**Figure 3 materials-14-04767-f003:**
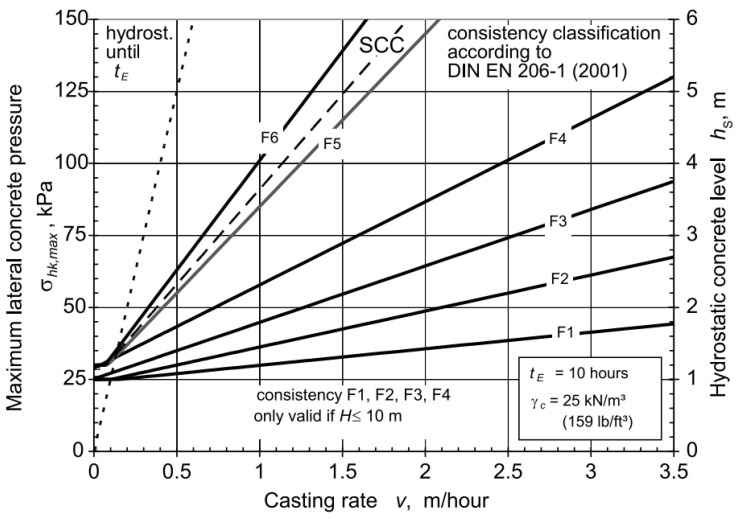
Maximum lateral pressure for concretes with a final setting time of 10 h [75] (reprinted with permission from © ACI).

**Figure 4 materials-14-04767-f004:**
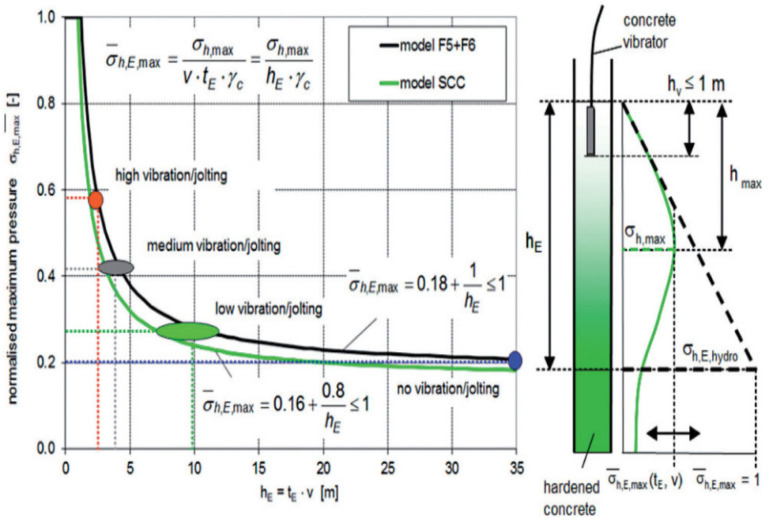
Normalized maximum pressure for high flowable concrete [42] (reprinted with permission from © John Wiley and Sons).

**Figure 5 materials-14-04767-f005:**
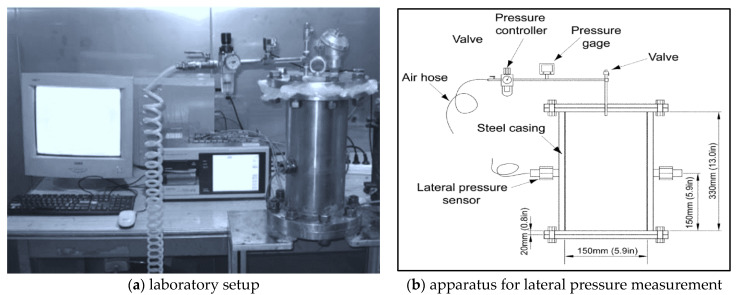
Laboratory setups for lateral pressure measurement (reprinted from [74], with permission © John Wiley and Sons).

**Figure 6 materials-14-04767-f006:**
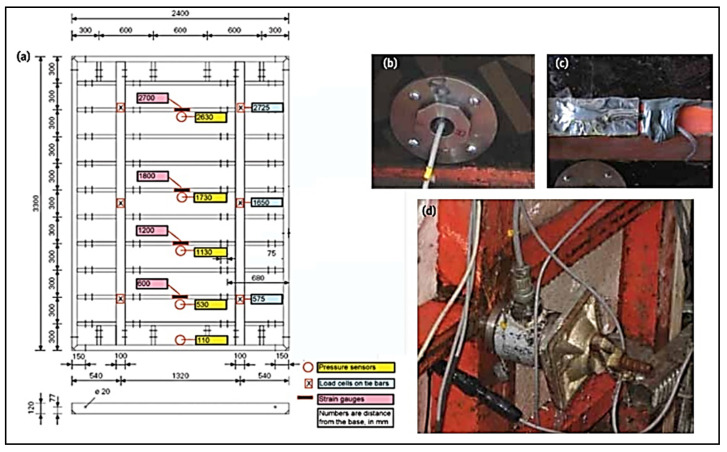
Full-scale experimental setup for pressure measurement: (**a**) location of sensors; (**b**) mounted pressure sensor; (**c**) strain gauge; (**d**) load cells [80] (reprinted with permission from © ACI).

**Figure 7 materials-14-04767-f007:**
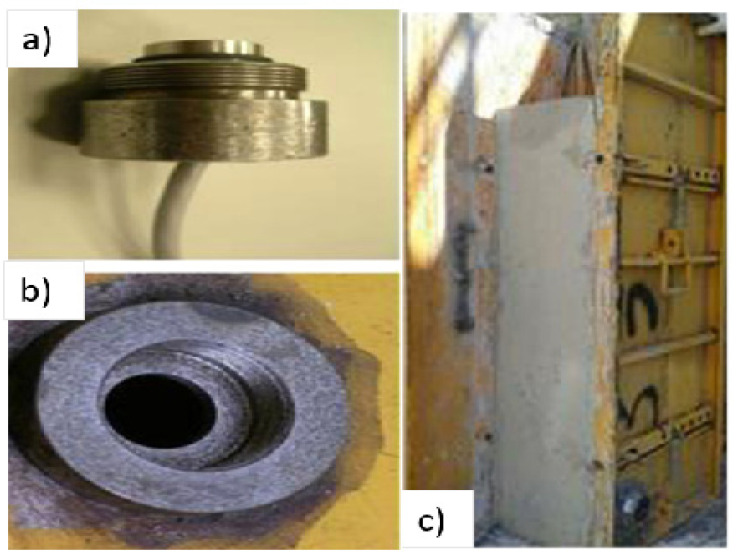
Laboratory setup with steel formwork (**a**) pressure sensor; (**b**) welded pressure diaphragm (**c**) formwork setup [40] (reprinted with permission from © 2021 Elsevier Ltd.).

**Figure 8 materials-14-04767-f008:**
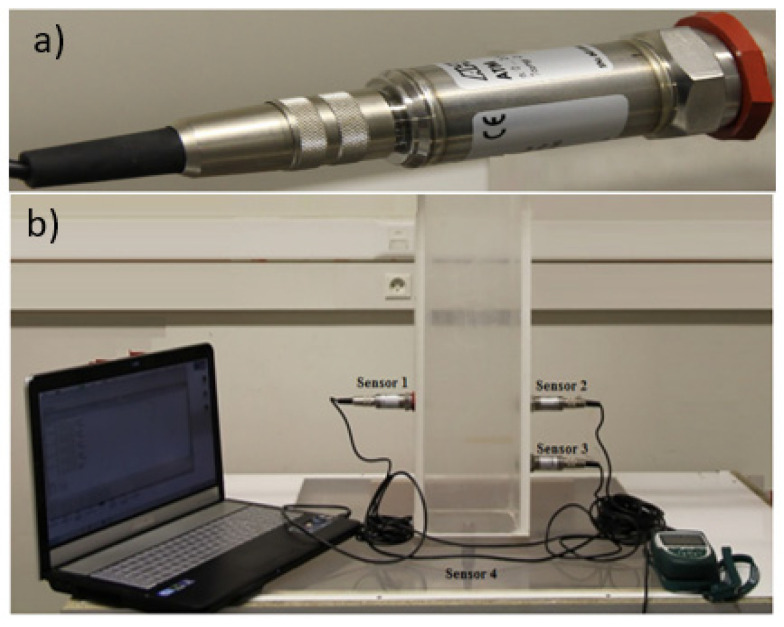
Laboratory setup. (**a**) Flush diaphragm sensor. (**b**) Laboratory setup for pressure monitoring [11] (reprinted with permission from © 2021 Elsevier Ltd.).

**Table 1 materials-14-04767-t001:** Factors Affecting Lateral Form Pressure Exerted by SCC.

Category	Parameters	References
Concrete Mix design	Gradation, shape, texture, and amount of fine and coarse aggregate	[1,12,24,25,26]
	Water to cement ratio	[21,24,27,28,29]
	Amount and type of SCMs, Amount and type of chemical admixtures	[1,12,24,30,31,32,33]
	Cement type and amount	[12,27,32,34,35,36,37,38,39]
Fresh concrete properties	Concrete temperature	[14,23,24,38,39,40]
	Setting time (rate of hardening)	[12,32,41]
	Concrete density	[1,41]
	The initial low shear stress	[17,24,36]
	Slump flow and T50 (consistency class)	[17,37,42,43,44]
	Thixotropy and viscosity	[12,36,42,43,44,45,46,47,48]
Placement technology	Casting rate and casting method	[13,23,24,36,48,49,50,51]
	Humidity and ambient temperature	[52,53]
	Reinforcement	[12,39,47,54,55]
	Pumping location	[37,47,55,56,57,58]
	Size of the structure, casting height	[12,36,59]
	Type of formwork and its geometry (including stiffness, surface friction, surface roughness, use of demoulding agents, weight)	[12,32,46,60,61,62,63,64,65]
	External stresses imposed by workers, equipment and materials, possible external loads created, e.g., by wind, pressure sensor location and mounting direction of the sensors	[12,60]

**Table 2 materials-14-04767-t002:** The influence of casting rate on the formwork pressure for different mix designs.

Casting Rate m/h	Approximate Max Recorded Pressure (kPa)	Associated Hydrostatic Pressure (kPa)	Height (m)	Reference
19	97	93	0.8	[37]
10	180	290	12.5	[67]
7	24.01	28.5	1.2	[17]
6	101	155.3	6.6	[68]
3.5	23	25	2.6	[70]
2.74	33.78	24.54	1.10	[51]

**Table 3 materials-14-04767-t003:** Form pressure variables included in the mathematical models.

Model Reference	Variables Included									
	Casting Rate	Concrete Temperature	Density	Formwork Geometry	Setting Time	Casting Height	Yield Stress	Maximum Size Aggregate	Slump Flow	Reinforcing Bar
Proske and Graubner [73]	X		X		X					
Kwon et al. [46]	X				X					
DIN 18,218 [75]	X	X	X		X	X				
Khayat and Omran [68]	X	X	X		X	X	X	X		
Graubner et al. [42]	X		X		X	X				
Gardner et al. [23]	X		X						X	
Teixeira et al. [76]	X	X	X	X		X			X	
Assaad and Matar [56]					X					X
Ovarlez and Roussel [4]	X		X	X		X	X			

**Table 4 materials-14-04767-t004:** Overview of the key findings from the literature.

Main Study Parameters	Key Findings	Reference(s)
Casting method, workability, and mix proportions.	1. Casting method influences the pressure amount and casting from the bottom generates high pressure at the bottom than casting from the top. 2. High slump flow affects lateral pressure. 3. Mix design influence the setting time having a subsequent influence on the pressure.	[37]
Formwork pressure, mixture proportions, height, casting rate.	A high casting rate induces high lateral pressure.	[34,46,60,74]
Pore water pressure, lateral pressure, time.	Form pressure diverges from hydrostatic pressure due to the thixotropic property of concrete.	[13]
Addition of mineral admixture and monitor the pressure change.	Mineral admixture such as processed clays lessens the lateral pressure.	[7]
Casting height, casting rate, the temperature of concrete and static and dynamic yield stress.	Lateral pressure exerted by SCC is less than the hydrostatic pressure.	[77]
Mix proportions, strain, formwork pressure, tie tension force.	Good correlation between lateral pressure and form deformation (strain) and a good correlation between the tie tension force and the pressure.	[70]
Slump flow and method of placement.	Pressure varies depending on the class consistency (slump flow) and method of placement.	[41]
Casting rate and slump flow.	Lateral form pressure depends on the performance of the admixture and placement rate.	[23]
Casting rate, slump loss, pressure, Wall geometries.	A notable correlation between casting rate, slump flow and the pressure were found in the study.	[68]
Casting rate.	A high casting rate leads to high pressure.	[19]
Viscosity, reinforcing rebar, casting location.	Pumping concrete from the bottom generates higher lateral pressure than from the top.	[46]
Recycled aggregate, Vertical reinforcement bars.	The finding indicated that using recycled aggregate reduces the initial pressure due to high surface roughness which increases the internal friction.	[52]

**Table 5 materials-14-04767-t005:** Formwork pressure measurement tools.

Pressure Measurement Tools	Formwork Type	Type of Structure\Dimensions	Reference
Mounted pressure sensor	Steel	Wall structure of dimensions 0.2 m × 0.75 m × 2.7 m & 0.25 m × 4.9 m × 4.7 m & 0.20 × 0.20 × 0.975 m	[37]
Strain gauge- based pressure sensors	Steel	Three walls and one column	[46,74]
Flush Diaphragm Millivolt Output Type pressure sensors	Steel	Retaining walls	[13]
Pressure transducers	Steel	Lab setup	[49]
Pressure sensors with 19 mm diameter and electronic transducers with 0–1380 kPa range and 0.25% accuracy	PVC	Lab setup using PVC with a diameter of 200 mm and a height of 700 mm.	[67]
Honeywell ABH100PSC1B pressure sensors rated for 0 to 689 kPa.	Thick plywood panels mounted on steel frames	Security Wall 0.27 m thick, 6 m tall, and 400 m long wall	[70]
Pressure transducers Omega PX43E0-100GI and load cells attached to tie-bars and pressure transducers installed in the inner surface of fresh concrete	Steel	Used 8 different walls dimensions Walls Nos. 1, 3, 5 and 7 are 6.6 m in height, 2.4 m in length, and 0.2 m in thickness. Walls Nos. 2, 4 and 6 differ only by height = 4.2 m. Wall no 8 is 4.2 m in height but had a thickness of 0.4 m	[68]
Pressure transducer	Steel	Column 2 m height	[19]
Pressure sensors with a diameter of 87 mm placed at 0.135 m, 0.375 m and 0.75 m from the bottom	N\A	Column (0.2 × 0.2 × 1.2) m	[17]
Linear variable differential transformers and high-precision digital micrometre strain gages	A Plexiglas acrylic	Lab setup rectangular sample 1600 mm height, 400 mm length, and 200 mm width	[79]
Flush diaphragm pressure sensors	Transparent plastic	Lab setup with square column dimension 16 × 16 × 70 cm	[11]

## Data Availability

Data sharing not applicable. No new data were created or analyzed in this article. Data sharing is not applicable to this article.

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
