# Peer review of "Lateral Formwork Pressure for Self-Compacting Concrete—A Review of Prediction Models and Monitoring Technologies"

_materials, 2021, doi:10.3390/ma14164767_

Round 1

Reviewer 1 Report

Dear authors,
thank you for such an extensive study of the current situation. As a review, it is clearly conceived and contains a lot of information.
Although the literature is extensive, it contains some articles with the same value of information. On the contrary, I recommend extending and recalling the results of these studies:
10.3390 / ma14061549
10.3390 / cryst10030220
If we take this article as a pure review there is not much to complain about.
The description of the models is adequate, the description of the results from the literature is adequate.

A few notes:
the reference list contains errors and is not according to the template.
the text contains US and UK terms - "behavior" vs "behaviour" etc.
the article contains a few bugs

Regards

Author Response

Reviewer Comment

Response

I recommend extending and recalling the results of these studies:
10.3390 / ma14061549
10.3390 / cryst10030220

We addressed the comment and discussed the finding from   ma14061549 but not cryst10030220 because the later is not serving much in the content of the manuscript.

A few notes: the reference list contains errors and is not according to the template.
the article contains a few bugs

The references are updated according to the journal template. Now it is free from errors.

the text contains US and UK terms - "behavior" vs "behaviour" etc

The comment addressed accordingly.

Reviewer 2 Report

Please follow file attached,

Regards.

Author Response

Reviewer Comment

Response

Introduction is short and covers just 23 references.

That is true, but we tried to avoid repeating the information that comes after.

Parameters, Affecting the Lateral Framework Pressure

A very welcome Table 1, which covers the factors affecting lateral form pressure exerted by SCC and

the references related. But the interrelationship with text presented below is a little chaotic and can cause

difficulties for the readers.

Some references (for ex. [64], [66], [70]) are mentioned in text (59), (90), (104) but not included in this

table in appropriate category. And opposite, there are references [1] included in Table 1, without or with

“one sentence” comments. Why?

The sequence of the references in this chapter is really violated.

Wouldn’t it be better to present Table 1 at the end of this chapter as a summary?

Although, after such a detailed table, no large comments are needed, as the reader could easily find

more detailed relevant information on the basis of the references provided

Table 1 has been updated according to the comment.

Yes, in the description below the table we stressed on the findings that serve good in delivering the idea, rewriting everything mentioned in the table will make it scattered. However, I addressed this comment.

The idea of placing the table at the opening of the subtopic is to make it easy for the readers. (in my opinion).

Modelling

It should be emphasized once again that presenting the results of the survey in tabular form (here and in

other sections) is the strong feature of the review article.

However, there are a lot technical inaccuracies in text:

What mean (N/kg) in 167? Is it really required?

Please balance the format of formulas 1, 2, 4, 6, 8, 9, 11, 12 (size, “*” “x”

Please balance the format of table 3

Figure 3. What about permission of authors?

226, 283 - Pmax

234 - Dmin

The height in some places “h” or “H”. It is necessary to unify.

235, 271, 272, 275, 307 indices PVτ0rest@15min and other...?

Figure 4 is not mentioned in the text?

366 Figure 7

The comments have been addressed accordingly.

Formwork pressure – monitoring and measurement

323 – Why in capital letters?

Edited

The list of references should be revised effectively, according to requirements of publisher:

 Publishing year should be in “Bold”;

 Surname, J. D, not opposite (everywhere);

 Authors in 483, 485, 535, 589, 592, 586, 601, 616……….. ?;

 Abbreviated Journals names in “Italic”;

 Book title should be in “Italic”;

 ………….

 Please follow instructions for authors more carefully.

The comment has been addressed; the format is updated accordingly.

Reviewer 3 Report

The presentation is very good. Figure 4 is not cited in the text.

Author Response

The presentation is very good. Figure 4 is not cited in the text

(Response to reviewer: The comments was addressed accordingly, see track changed DOC).

Reviewer 4 Report

The authors in the review paper described the modeling and measurement of Lateral Formwork Pressure for Self-Compacting Concrete. Models developed over the years 2010-2020 were analyzed. The authors did not consider previous models, eg Proske, Vanhove (2004) or Roussel Ans Ovartez (2006) and also important for reflection models developed, for example by Gardner (2014) or American Concrete Institute (ACI) guidelines, 347. In the review, the results were not included in the review Research published in recent years 2019-2020. Work does not include the latest testing in the field of Lateral Formwork Pressure measurements, eg Materials 2021, 14, 1549. https://doi.org/10.3390/MA14061549. Overview should be based on a browser of a larger number of works published in a wide time aspect. That's why paper requires significant additions. 

Author Response

Reviewer Comment

Response

The authors in the review paper described the modeling and measurement of Lateral Formwork Pressure for Self-Compacting Concrete. Models developed over the years 2010-2020 were analyzed. The authors did not consider previous models, eg Proske, Vanhove (2004) or Roussel Ans Ovartez (2006) and also important for reflection models developed, for example by Gardner (2014) or American Concrete Institute (ACI) guidelines, 347. In the review, the results were not included in the review Research published in recent years 2019-2020. Work does not include the latest testing in the field of Lateral Formwork Pressure measurements, eg Materials 2021, 14, 1549. https://doi.org/10.3390/MA14061549. Overview should be based on a browser of a larger number of works published in a wide time aspect. That's why paper requires significant additions.

Thank you for the comments.

Regarding the models, Proske & Graubner has been discussed in the article. However, Vanhove et al. (2004) in the article entitled

Prediction of the lateral pressure exerted by

self-compacting concrete on formwork adopted Janssen’s model however the outcome according to the conclusion in the paper says “Janssen’s model tends to underestimate the material’s lateral pressure. The model overestimates internal friction and friction on the walls, which explains the need to introduce a coefficient that takes into account the inequality”. Hence, we believe there was no need to discuss it in the article as a solid model to predict the pressure.

Concerning Roussel and Ovarlez (2006), we have added that accordingly.

But for American Concrete Institute (ACI) guidelines, 347 model is basically for normal vibrated concrete and doesn’t fit for SCC.

We have also discussed the latest findings presented by the research “Materials 2021, 14, 1549”. https://doi.org/10.3390/MA14061549.

Reviewer 5 Report

Dear Authors,

please see the attached file.

Author Response

Reviewer Comment

Response

Chapter 2 - The presentation of the  parameters affecting the lateral formwork could be subdivided in three paragraphs (from 2.1 to 2.3.) according to their category (i.e., concrete mix design, fresh concrete properties, placement technology). The reviewer believes that it could improve readability of the section.

The comment arrangement was restructure according to the reviewer comment

Line 64 - The effect of water to cement ratio on the lateral pressure could be further discussed.

More discussion has been added

Line 97 -The acronym "VMA" is not previously introduced and should be clarified.

Added

Line 108 - "Figure 1 demonstrates the relationship between flowability and pressure and shows that high flowable concrete generates high lateral pressure". According to Figure 1, this is not true for all the cited authors (e.g. for Benaicha et al., 2019). This could be better explained and discussed.

The justification is added and that could be due to the casting rate, geometry of the form used in the experiments and sensors accuracy. In fact, if reinforcement is used then there will be some blockage at the sensor diaphragm causing inaccurate pressure reading.

Figure 1-The number of the reference (reported at the end of the document) is missing when citing the works in the graphs. Moreover, it is not completely clear if the x-axis presents the same scale for both the slump flow and the maximum lateral pressure.

1)       

The figure was updated accordingly.

Line 127 - "As observed from table 2 all the recorded maximum pressure is less than hydrostatic pressure except for casting rate 2.74 m/h because revibrating was applied after casting". According to the table, this does not apply for a casting rate of 2.74 m/ h. This should be highlighted and further discussed.

Discussion was added to clarify this point.

Line 158 - The acronym "PVD" is not previously introduced and should be clarified.

It is PVC, type error Polyvinyl chloride, and it has been mentioned in previous section.

correction was made.

Line 166 - The reported equation should be numbered as Eq. 1 for ease of understanding.

The template of the journal uses only numbers without Eq. However, we updated that while reporting in the text.

Line 184-The neglected factors could be recalled at this point of the paper.

The conclusion demonstrates that and recalled the neglected factors.

Line 194 - It is not completely clear the meaning of the symbol "tb".

Clarified.

Eq. 6 - It is not completely clear the meaning of the symbol "m" in the equation.

Referring to the original document, it doesn’t say anything, I deleted it. Because it signifies nothing.  

Figure 6 a)- The readability of the picture should be improved.

Improved accordingly

Lines 353, 447 -A   typo is present in the line

Addressed

Round 2

Reviewer 4 Report

Autorzy zmienili manuskrypt. Artykuł można opublikować w aterials w tej wersji.